# Hormonal Regulation of Renal Fibrosis

**DOI:** 10.3390/life12050737

**Published:** 2022-05-16

**Authors:** Polina A. Abramicheva, Egor Y. Plotnikov

**Affiliations:** 1Belozersky Institute of Physico-Chemical Biology, Lomonosov Moscow State University, 119991 Moscow, Russia; abramicheva.polina@belozersky.msu.ru; 2Kulakov National Medical Research Center of Obstetrics, Gynecology and Perinatology, 117997 Moscow, Russia

**Keywords:** fibrosis, fibroblasts, kidney, oxidative stress, water–salt balance, sex hormones

## Abstract

Fibrosis is a severe complication of many acute and chronic kidney pathologies. According to current concepts, an imbalance in the synthesis and degradation of the extracellular matrix by fibroblasts is considered the key cause of the induction and progression of fibrosis. Nevertheless, inflammation associated with the damage of tissue cells is among the factors promoting this pathological process. Most of the mechanisms accompanying fibrosis development are controlled by various hormones, which makes humoral regulation an attractive target for therapeutic intervention. In this vein, it is particularly interesting that the kidney is the source of many hormones, while other hormones regulate renal functions. The normal kidney physiology and pathogenesis of many kidney diseases are sex-dependent and thus modulated by sex hormones. Therefore, when choosing therapy, it is necessary to focus on the sex-associated characteristics of kidney functioning. In this review, we considered renal fibrosis from the point of view of vasoactive and reproductive hormone imbalance. The hormonal therapy possibilities for the treatment or prevention of kidney fibrosis are also discussed.

## 1. Introduction

Fibrosis is a pathological process of the extracellular matrix (ECM) deposition, leading to organ dysfunction or a complete loss of functional activity. Fibrosis development is a characteristic of all forms of chronic kidney disease (CKD). The most common causes of CKD in developed countries are type 2 diabetes mellitus and ischemic and hypertensive nephropathy, which often accompanies other diseases. CKD is often associated with acute kidney injury (AKI): although both conditions are diagnosed as separate diseases, CKD can develop as a result of untreated AKI. Well-known biomarkers of the early stage of AKI, kidney injury molecule (KIM-1) and neutrophil gelatinase-associated lipocalin (NGAL), are currently considered potential markers for AKI-to-CKD transition [1]. As CKD progresses, there is excessive production of ECM by fibroblasts enriched with type I and III collagen and vessels basement membrane components (type IV and V collagen, fibronectin, laminin, perlecan and heparan) in peritubular space. These processes lead to tubular fibrosis, glomerular sclerosis and renal artery stenosis [2,3,4]. Fibrogenesis in CKD is a multi-stage process. Persistent renal tissue injury and nephron cell death cause an acute inflammatory response associated with an infiltration of activated neutrophils, macrophages and dendritic cells. Further, chemokines and cytokines secreted by these cells in the lesion induce fibroblasts transformation toward myofibroblasts that synthesize ECM, where excessive accumulation leads to apoptosis and tubular atrophy, disruption of kidney tissue architecture and, ultimately, functional kidney failure [5,6,7].

Currently, several sources of activated myofibroblasts in the kidney are considered: resident interstitial fibroblasts, which become myofibroblasts as a result of stimulation [8,9,10,11]; fibroblasts from red bone marrow; epithelial cells of the renal tubules, which undergoes epithelial-to-mesenchymal transition (EMT) into mesenchymal myofibroblasts migrating to the adjacent interstitial parenchyma; endothelial cells; pericytes [12,13].

The kidney is a highly vascularized organ since it consumes a large amount of energy, and its metabolic activity, including maintaining the water–salt balance, is possible only with an adequate blood supply. Hormones are among key players in the regulation of blood supply, vascular tone and water and salt homeostasis in the kidney. By taking into account the broad role of hormones in kidney heath and disease, in this review, we considered renal fibrosis as a pathological process associated with hormonal imbalance.

## 2. Key Regulators of Kidney Fibrosis

### 2.1. Hypoxia and Mitochondrial Damage in CKD

Fibrogenesis is triggered in response to a chronic inflammatory process, which could be associated with impaired blood flow, and as a result, insufficient oxygen supply. Hypoxia is an important factor associated with the development of interstitial fibrosis in CKD, as the kidney is extremely sensitive to acute and chronic ischemic injury. Hypoxia is directly related to impaired functioning of mitochondria provided aerobic cell metabolism. Ischemic tissues switch from aerobic metabolism to glycolysis. The glycolytic pathway may be a critical point in the regulation of collagen synthesis and the development of fibrosis in various pathologies, including CKD. Enhanced glycolytic flux alone cannot meet the high metabolic demands of fibroblasts, so increased carbon supply through alternative pathways (e.g., glutamate metabolism) is needed to support biosynthetic requirements in fibrosis [14]. Moreover, glutaminolysis is required for transforming growth factor β1 (TGFβ1)-induced myofibroblast differentiation and activation [15]. The ratio of different pathways of cellular energy supply also plays an important role; for example, suppression of the fatty acid oxidation pathway leads to an increase in the synthesis of ECM proteins even with unchanged glycolytic activity [12,16].

Under conditions of ischemia, mitochondria become a source of pathological amounts of reactive oxygen species (ROS), causing oxidative stress. It is well known that a variety of renal pathologies (AKI, CKD, nephrotic syndrome and metabolic disorders) triggers ROS production, although the mechanisms of such induction may be different. Oxidative stress plays an important role in the development of tubulointerstitial fibrosis (TF) and glomerulosclerosis. There are a number of signaling molecules and cascades associated with the development of oxidative stress in the kidney: NADPH oxidase (Nox), nuclear factor erythroid-2 related factor 2 (Nrf2) and the peroxisome proliferator-activated receptor γ (PPARγ) signaling [17]. It is known that Nox1, Nox2 and Nox4 producing ROS play a central role in oxidative stress in CKD and contribute to the development of fibrosis, and their impact on this process is ambiguous. There is evidence that Nox4 promotes the transformation of fibroblasts into myofibroblasts in vitro [18], while in vivo, Nox4 has a nephroprotective effect in a model of unilateral ureteral obstruction (UUO): Nox4-knockout mice after obstruction develop TF and oxidative stress faster and the expression of the antioxidant Nrf2 protein decreases [19,20]. Nrf2 is a transcription factor that regulates the expression of a number of genes that resist inflammatory and oxidative damage, including genes encoding thiol-associated proteins, detoxifying enzymes, antioxidant and stress response proteins [17]. Growing data indicate that Nrf2 signaling has a protective function in various models of CKD and appears to be associated with the suppression of fibrogenesis. For example, dihydroquercetin demonstrated antifibrotic effects and was shown to activate the Nrf2 signaling pathway in the UUO model in mice [21]. Similarly, treatment with transcription factor PPARγ agonists (for example, pioglitazone) suppresses TGFβ1 signaling, TF progression and ROS generation [17,22,23].

In some studies, mitochondrial dysfunction is assumed as an important factor in renal fibrosis development [24,25,26]. There are several mitochondrial-associated processes that undergo major changes in CKD. These processes and some of their key participants are described below:Mitochondrial dynamics. Excessive phosphorylation of GTPase dynamin-related protein 1 Drp1 leads to increased mitochondrial fragmentation, increased fibroblast proliferation and provocation of renal fibrosis [27];Biosynthetic processes. In CKD, decreased expression of one of the key regulators of mitochondrial DNA transcription, PPARγ-coactivator 1α (PGC1α), is observed. A decrease in the rate of renal fibrosis progression and mitochondrial structure restoration in mouse models of PGC1α overexpression was shown [28];Mitochondrial signaling. Impaired signaling involving protein kinase Akt leads to dysregulation of ATP production, caspase activation, subsequent apoptosis and fibrosis development [29];Endoplasmic reticulum. Disturbance in mitochondria–endoplasmic reticulum contacts leads to inflammation and fibrosis development, as well as AKI-to-CKD transition. Activation of the endoplasmic reticulum signaling pathway associated with protein folding (unfolded protein response pathway) can trigger apoptosis. Several animal models of AKI showed stimulation of activating transcription factor 6 ATF6 that regulates unfolded protein response pathway, leading to impaired fatty acid β-oxidation by suppressing PPARα expression and ultimately inducing tubular inflammation and fibrosis [3,30].

Thus, mitochondrial dysfunction and associated oxidative stress can be called one of the non-classical causes of kidney fibrosis.

### 2.2. Matrix Metalloproteinases in Kidney Fibrosis

Matrix metalloproteinases (MMPs) are zinc-containing endopeptidases with a wide range of substrate specificities. They are secreted as precursors by the renal tubular epithelium and internal glomerular cells in response to various stimuli: oxidative stress, UV radiation, cytokines, etc. [4,31,32,33,34]. The activity of MMPs is regulated by a family of tissue inhibitors of metalloproteinases (TIMPs), which have a dual dose-dependent effect on the enzymes. On the one hand, TIMPs suppress the activity of mature MMPs, and on the other hand, they activate MMPs’ pro-enzymes and stimulate mature enzymes at low concentrations. MMPs are involved in kidney development regulation and the pathogenesis of renal diseases; thereby, a balance between MMPs’ levels and their inhibitors is needed to maintain renal homeostasis.

MMP-7 (matrilysin-1) is considered a key regulator of kidney fibrosis pathogenesis, affecting EMT, TGFβ1 signals, ECM deposition, apoptosis and proliferation of renal tubular epithelium [4,35]. This enzyme is almost not expressed in a healthy kidney, but its expression significantly increases in AKI and CKD. In addition to degradation of ECM components, MMP-7 also cleaves a wide range of substrates, such as E-cadherin, syndecan, Fas-ligand, nephrin, plasminogen, as well as pro-enzymes MMP-2 and 9 [4,34]. The function of MMP-7 in kidney disease is complex and pathology-dependent. On the one hand, it protects renal tubular cells from AKI, providing pro-survival and regenerative signals, but on the other hand, it promotes fibrosis and progression of CKD [36]. Loss of MMP-7 was associated with an increased expression of several proinflammatory cytokines, which regulate various aspects of inflammation [35]. It is assumed that serum MMP-7 levels might serve as a noninvasive prognostic biomarker in IgA nephropathy patients [37]. MMP-7 expression is regulated by various signaling pathways, including Wnt/β-catenin signaling, which provides the communication of fibroblasts, and is inactive in a healthy kidney. Wnt activation leads to stabilization of β-catenin, its translocation to the nucleus followed by binding to the T-cell factor TCF (lymphoid enhancer-binding factor and stimulation of target gene transcription), EMT stimulation and MMP-7 expression and secretion. In turn, released MMP-7 induces FasL expression in interstitial fibroblasts, a key participant in the receptor-mediated apoptotic signaling pathway, and triggers apoptosis in these cells [4].

MMP-2 and MMP-9 (known as gelatinases) are produced in small amounts by mesangial cells and renal epithelium. These enzymes cleave the denatured collagens (gelatins) and laminin, as well as some chemokines [38]. They also play an important role in the ECM degradation and participate in the development of CKD [39]. Factors stimulating the expression of these enzymes include TGFβ1/Smad signaling, p38 MAPK (mitogen-activated protein kinase), Notch and hypoxia. As with the case of MMP-7, there is a close relationship between the activation of MMP-2 and MMP-9 and the signaling of several growth factors. MMP-2 stimulates the expression of fibroblast growth factor, and MMP-9 has a similar effect on TGFβ1; both lead to kidney fibrosis progression [39]. MMP-2 and MMP-9 are two of the proteinases that have been widely studied in human diabetic kidney disease. The concentrations and the activity of both enzymes are increased in the urine of type 1 and 2 diabetes patients [38]. Clinical studies of diabetic nephropathy and kidney transplantation outcomes showed that M2-macrophages localized at the fibrotic areas and actively produced MMP-2/9 [40].

Thus, one of the reasons for kidney fibrosis development is an imbalance in MMPs/TIMPs ratio, and this balance can be a potential target for hormone regulation.

### 2.3. The Role of TGFβ1 in the Regulation of Kidney Fibrosis

Despite the large number of reviews devoted to the role of TGFβ1 in fibrosis, it is necessary to mention renal fibrosis regulation by this growth factor since many hormones act on the kidney through TGFβ1-signaling. TGFβ1 is a growth factor with a pleiotropic effect that plays an important role in cell proliferation and differentiation, induces the synthesis of ECM components and is the main cytokine involved in kidney fibrosis development [4,41,42,43]. The mammalian TGFβ1 family is represented by three isoforms: TGFβ1, TGFβ2 and TGFβ3. In humans, TGFβ1 is secreted as an inactivated high molecular weight complex. It is the main regulator of tissue healing after tissue damage caused by various diseases. However, in the case of its overproduction, excessive connective tissue formation occurs, and this provokes surrounding parenchyma damage, collagen deposition and renal sclerosis.

TGFβ1 stimulates two transmembrane receptors (serine-threonine kinases I and II), which activate two intracellular mediators, Smad2 and Smad3, through phosphorylation. Further, these mediators combine with each other and with a third mediator, Smad4. This ternary complex is transported to the cell nucleus, where it regulates the expression of genes encoding proteins required for EMT. Integrin-linked kinase (ILK) acts in addition to the signaling pathway described above to complete the EMT process. ILK interacts with integrins and other cytoskeletal proteins that control a number of integrin-dependent processes: cell adhesion, cell reshaping, gene expression and ECM accumulation. ILK stimulation leads to the loss of epithelial E-cadherin, increased expression of fibronectin and its extracellular accumulation, stimulation of MMP-2 expression and excretion, and increased cell migration. Thus, the TGF-β1/Smad and ILK signaling pathways complement each other, creating a complex pathway where Smad second messengers play an important role, and EMT is the outcome [12]. TGFβ1 stimulates EMT in the kidney, increasing the production of α-smooth muscle actin and vimentin by increasing the expression of connective tissue growth factor (CTGF) and through Smad2/3 phosphorylation. CTGF is not expressed in a healthy kidney, and its level correlates with fibrosis progression. TGFβ1 also induces the synthesis of ECM proteins: type I collagen and tenascin-C [13].

A close relationship was demonstrated between MMP-7 signaling (Wnt/β-catenin) and TGFβ1/Smad4 signaling. Sirtuin I, a class III NAD+-dependent deacetylase that deacetylates both histone and non-histone proteins, demonstrated nephroprotective effects in some models of CKD [44] since it suppresses MMP-7 protein expression by Smad4 deacetylation [4].

Activation of TGFβ1 signaling is associated with oxidative stress development since TGFβ1 activates a number of Nox [17]. TGFβ1 also contributes to the disruption of mitochondrial function by inducing their fragmentation, which was shown in renal tubular cells in mice with CKD [45]. However, the molecular mechanisms mediating mitochondrial fragmentation in response to TGFβ1 remain unknown.

Despite the well-known profibrotic effect of TGFβ1, blocking its signaling and expression does not always cause a nephroprotective effect. For example, in a UUO model on keratinocytes with overexpression of inactive TGFβ1, a decrease in the rate of renal fibrosis was shown [46], as well as in the case of Smad7 overexpression, which inhibits TGFβ1 signaling by a negative feedback mechanism [47,48].

Thus, the question of how to reduce or reverse kidney fibrosis by blocking the synthesis, secretion and signaling of TGFβ1 remains open.

## 3. Hormones and Fibrosis

Most of the hormones discussed in this review affect the progression of fibrosis indirectly through TGFβ1 stimulation. However, some hormones are able to directly stimulate fibrogenesis in the kidney.

### 3.1. Angiotensin II

Angiotensin II (Ang II) is the major effector of the renin–angiotensin–aldosterone system (RAAS), the main function of which is to maintain optimal blood pressure. Ang II is formed from a precursor protein, angiotensinogen, synthesized in the liver. Angiotensinogen is cleaved to angiotensin I by renal renin protease. In turn, angiotensin I is processed by the angiotensin-converting enzyme (ACE), which cleaves Ang I to the Ang II octapeptide. Ang II possesses two key functions: it causes vasoconstriction and stimulates the secretion of the mineralocorticoid aldosterone (Aldo), which enhances sodium reabsorption by retaining water in the kidney [49]. In the kidney, this vasoactive peptide activates mesangial and tubular cells and increases the synthesis of ECM proteins by interstitial fibroblasts. In the case of proteinuria under conditions of kidney damage, pathological activation of the RAAS occurs, which causes activation of ECM synthesis and suppression of its degradation; thus, it triggers fibrosis development. TGFβ1 is an important mediator of the effects of Ang II on TF development [13]. Ang II directly stimulates the release of TGFβ1 from the latent complex and the expression of the TGFβ1 receptor type II serine–threonine kinase gene through its type 1 receptors (AT1). Moreover, it indirectly affects TGFβ1 signaling through monocyte chemoattractant protein 1, endothelin-1 and osteopontin, activating a number of signaling systems and transcription factors: activating protein-1, signal transducer and activator of transcription (STAT) and cAMP response element-binding protein (CREB) family factors, nuclear factor kappa-light-chain-enhancer of activated B cells (NFκB) [50,51]. In a rat model of obstructive nephropathy, TF progression was accompanied by an increase in STAT3 phosphorylation and the expression of proapoptotic Bax and Bcl-2 genes, while the administration of the AT1 antagonist losartan abolished these effects [52]. Losartan is also known to suppress EMT in the kidney [53].

Ang II also stimulates renal fibrosis in a TGFβ1 signaling-independent manner. It is known that Ang II can directly induce the expression of CTGF, a downstream effector of TGFβ1, by stimulating EMT [51]. Ang II can induce inflammation and fibrosis of the kidney by directly binding to myeloid differentiation protein 2, an accessory protein of the Toll-like receptor 4 (TLR4), leading to the activation of NF-κB, ERK kinase, proinflammatory and profibrotic molecules [54].

The influence of Ang II on the late stages of fibrogenesis may be associated with the ECM accumulation not by stimulation of the expression of its components but by the suppression of its degradation. Thus, in a culture of bovine aortic endothelium and human adipocytes, Ang II induces the expression of the inhibitor of tissue plasminogen activator 1 (PAI-1), which suppresses fibrinolysis, thereby provoking both fibrosis and thrombosis [55,56,57].

Ang II contributes greatly to the development of oxidative stress in the kidney through AT1 activation, regulating the expression and activity of Nox1 and superoxide dismutase (SOD). This effect disappeared when the AT1 antagonist candesartan was administered [58].

In contrast to the proinflammatory, prooxidant, profibrotic and prohypertensive effects of AT1 receptor signals, AT2 activation induces anti-inflammatory, antioxidant, antifibrotic and antihypertensive responses [59,60]. Ang II through AT2 implements a nephroprotective effect and suppresses fibrosis, which is associated with cyclic guanosine monophosphate (cGMP) activation and NO synthesis [50].

The kidney contains all the components of the RAAS and locally produces renal Ang II. The local RAAS of the kidney, in addition to the classical components, includes a number of other peptides and enzymes. Ang I and Ang II can undergo proteolytic cleavage by these enzymes to form other metabolically active angiotensin peptides that have their own specific receptors. The most known non-canonical participants in the local RAAS are angiotensin-converting enzyme 2 (ACE2) and neprilysin. Angiotensin 1-7 (Ang 1-7) heptapeptide is generated from Ang I and Ang II by these enzymes. Ang 1-7 can then be cleaved to another angiotensin peptide, alamandin. Ang 1-7 and alamandin have their own specific receptors, MAS1 and MrgD, which demonstrate effects similar to those of Ang II through the AT2 receptor—anti-inflammatory, antifibrotic and vasodilating [61,62]. Ang 1-7 suppresses the development of oxidative stress in CKD models by inhibiting Nox4 expression [63] and increasing catalase activity in the kidneys [64].

### 3.2. Aldosterone

Aldo is a mineralocorticoid hormone that is part of the RAAS. Its main effects on the kidney are the maintenance of water–salt balance in the distal nephron tubules and regulation of blood pressure. This hormone is also considered to be an important mediator of fibrosis in a number of organs, including the kidney [65]. It was shown that chronic Aldo administration causes TF and glomerulosclerosis [66], and its effects under conditions of TF are similar to those of Ang II. In CKD, Aldo acts on infiltrating inflammatory cells; inducing ROS production; stimulating epidermal growth factor receptor and AT1 expression; activating transcription factors NFκB and activating protein-1; increasing the release of proinflammatory cytokine TNFα and chemokines CCL2 and CCL5; and promoting TGFβ1 and PAI-1 production [66,67]. In a rat model of UUO, the fibrosis progression was reduced by the administration of the mineralocorticoid receptor (MR) antagonist spironolactone [68].

Incubation of HK-2 renal cell line with Aldo triggered EMT, as evidenced by E-cadherin loss, de novo expression of α-smooth muscle actin, and this process was completely blocked by the selective MR antagonist eplerenone. Aldo induces EMT through increased ROS production, and this production is mediated by mitochondria since it is blocked by rotenone but not by the Nox inhibitor apomycin [65,69]. Aldo administration removed the nephroprotective effect of ACE inhibitors in vivo in rats with remnant renal hypertension [70,71] and in spontaneously hypertensive rats [71]. Aldo is known to cause kidney damage and, as a result, mitochondrial dysfunction and oxidative stress by suppressing PGC1α expression [72]. Several authors consider that mitochondrial dysfunction is one of the first events during podocyte damage caused by Aldo [73]. It was shown that the use of the SOD mimetic MnTBAP in vitro and in vivo suppresses the activation of the Nod-like receptor pyrin domain containing 3 inflammasome and thereby alleviates Aldo-induced mitochondrial dysfunction [74].

### 3.3. Natriuretic Peptides

Natriuretic peptides are RAAS antagonists in the regulation of water–salt metabolism, and there is evidence of their participation in the development of kidney fibrosis. Atrial natriuretic peptide (ANP) is predominantly secreted by cardiomyocytes in response to the right atrium stretch. Brain natriuretic peptide (BNP) is released in response to afterload (myocardial stretch in systole), and C-type natriuretic peptide C (CNP) is secreted by endothelial cells in response to a variety of stimuli, including shear stress and production of proinflammatory cytokines [75]. The aforementioned peptides are vasodilators; they increase natriuresis and diuresis and reduce the production of RAAS components, the volume of circulating blood and blood pressure [76,77]. All of the mentioned peptides realize their action through receptor guanylate cyclases, activating cGMP-dependent protein kinase G (PKG), phosphodiesterases and ion channels, which ultimately causes natriuretic and nephroprotective responses [77]. cGMP is a key secondary messenger in ANP, BNP and CNP signaling. It is known that cGMP induces mitochondrial biogenesis in vivo and in vitro, which is important for acute and chronic kidney damage and characterized by mitochondrial dysfunction [78]. Increased renal fibrosis was observed in mice with a knockout for membrane guanylate cyclase A, indicating the antifibrotic effect of cGMP [79,80].

ANP is the main peptide in the systemic circulation. Recently it was discovered that the kidney contained the full biosynthetic apparatus for the production of ANP and enzymes that degrade it; it was suggested that ANP is involved in the pathogenesis of CKD [81]. Penna Della et al. showed that a high-salt diet caused an increase in ANP, TGFβ1 and hypoxia-inducible factor 1 alpha (HIF1α) expression in the same nephron segments: the glomerulus, the thick part of the ascending loop of Henle and the collecting duct. Thus, an increase in the expression of ANP and HIF1α may be part of an adaptive response to the profibrotic effect of TGFβ1 and oxidative stress caused by a high-salt diet. In addition, administration of TEMPOL (a SOD mimetic) was found to decrease ANP and HIF1α expression in the kidney [82]. It was previously demonstrated that activation of ANP/cGMP/PKG signaling phosphorylated Smad3 and abolished TGFβ1-induced pSmad3 nuclear translocation and later events, including EMT, rat pulmonary artery smooth muscle cell proliferation, and expression of ECM components [83] in mouse cardiofibroblasts [84]. In addition to being an antioxidant, ANP has anti-inflammatory effects, reducing NFκB and TNFα expression and the production of a number of chemokines and cytokines [85,86]. It is assumed that the protective effect of ANP on mitochondria is mediated by cGMP, preventing mitochondrial permeability transition and reducing membrane potential loss and cell death [87,88,89]. Despite these physiologically significant effects, the data linking the activity of ANP and cGMP signaling to mitochondrial function or related cascades are absent [78].

One way of ANP degradation in the bloodstream is cleavage by membrane metalloendopeptidase neprilysin, which is expressed in many tissues, including the kidney. Neprilysin also cleaves bradykinin and angiotensin 1-7, which have a natriuretic effect [81,90]. Thus, neprilysin regulates the pool of active hormones by proteolytic degradation of natriuretic peptides [78].

BNP is a well-known biomarker for heart failure, hypertension and cardiac hypertrophy; however, it is also elevated in CKD patients without cardiovascular disorders. Similar to ANP, BNP demonstrated a nephroprotective effect by inhibiting sodium reabsorption in the proximal and distal nephron, increasing the glomerular filtration rate by relaxing glomerular mesangial cells and inhibiting the action of a number of vasoconstrictors, including RAAS components [91,92,93]. BNP demonstrated antifibrotic activity in a number of organ systems, including the kidney. In the glomerulosclerosis model in vitro, BNP was shown to reduce ECM accumulation by podocytes incubated with TGFβ1, reduce TIMP2 expression and suppress glycogen synthase kinase 3 beta activity, which is closely associated with the pathogenesis of glomerulosclerosis [94] and is considered a key therapeutic target that determines the development of oxidative stress [95].

CNP is secreted by endothelial cells and, apart from natriuretic and vasodilatory effects, plays an important role in tissue remodeling [96,97]. Using human umbilical vein endothelial cells HUVEC, it was shown that CNP suppressed the expression of adhesion molecules ICAM-1, E-selectin and TNFα-induced NFκB activation [85]. The role of CNP in fibrosis remains unexplored; nevertheless, in a model of UUO in rats, CNP infusion reduced the deposition of type IV collagen in the kidney by suppressing the expression of TIMP1 and TIMP2 [97]. Sangaralingham et al. proposed using CNP as a biomarker of kidney aging and fibrosis development in this organ; they showed that in old rats, CNP levels in the blood plasma are greatly reduced, and their urine excretion dramatically increased [98].

### 3.4. Prolactin

Prolactin (Prl) is a 23 kDa polypeptide hormone with multiple targets and physiological functions. In the context of kidney physiology, Prl can be considered as a link between the hormones of the reproductive system and natriuretic peptides since the functions of prolactin are very diverse: reproduction, lactation, immunomodulatory action and regulation of water–salt balance. In the kidney, prolactin receptors are highly expressed in the glomeruli and proximal tubules. After filtration in the glomeruli, Prl also manifests its activity in other segments of the nephron: in the thick ascending part of the Henle’s loop, the distal tubule and the collecting duct. Prl has a natriuretic effect in the kidney, regulating the functioning of such sodium transporters as Na+/K+-ATPase, NCC, ENaC and NBCe1 [99,100,101,102]. There is a significant increase in the concentration of serum Prl in kidney diseases [103,104]; therefore, a number of authors refer to Prl as uremic toxins due to the relationship between its serum levels and uremia progression in CKD. This hormone accumulates in the blood along with loss of renal function, and its high levels are associated with worse cardiovascular outcomes [105]. Hyperprolactinemia is associated with increased Prl secretion and a decrease in its clearance due to CKD [81,106].

An endothelium is one of the targets of Prl action in the kidney. A cohort study showed that an increase in serum Prl concentration raised the risk of cardiovascular disorders in CKD patients without hemodialysis, as well as high mortality associated with cardiovascular dysfunction in patients on hemodialysis [107]. Conversely, Prl is reported to cause such positive effects as vasodilation and increased vascular permeability, and it stimulates angiogenesis [108,109].

Prolactin undergoes proteolytic degradation by several metalloproteinases and the cathepsin D enzyme with the formation of vasoinhibins that act on the endothelium and, in fact, are antagonists to prolactin [108,109]. Unfortunately, the role of these short peptides in the kidney is still poorly understood. It can only be assumed that vasoinhibins provoke CKD and TF development indirectly since they cause endothelial dysfunction, which ultimately leads to oxidative stress and the development of inflammation in the kidney. In bovine capillary endothelial cells, vasoinhibins were shown to stimulate the activation of PAI-1, which is associated with the development of fibrosis [110]. Furthermore, there was evidence that cathepsin D stimulated kidney fibrosis in a mouse model of CKD [111]; however, the enzyme deficiency increased the risk of AKI [112].

The scheme of the influence of vasoactive hormones on fibrosis development is shown in Figure 1.

### 3.5. Sex Hormones

#### 3.5.1. Testosterone

There are significant gender differences in the tolerance of various organs to damage. Gender is also a determining factor in the development of CKD. It is considered that women are more likely to suffer from CKD, but the rate of progression of this disease is higher in men [108]. It is interesting to note that such a physiological state as pregnancy protected the kidney from ischemic damage and reduced the rate of development of renal tissue fibrosis in the ischemia-reperfusion model in rats [113]. As for other kidney diseases, nephrolithiasis is known to be 2–3 times more common in men than in women [114]. The incidence of renal hypertension and diabetic nephropathy in men is higher than in women of the same age before menopause [115,116].

The effects of testosterone on the normal physiology and pathophysiology of the kidney are controversial and depend on the effective concentrations of this hormone. Testosterone administration at low doses to male rats demonstrated a nephroprotective effect during ischemia-reperfusion due to an increase in the TNFα/interleukine-10 ratio and a decrease in T-cell infiltration [117]. In a rat model of age-related kidney fibrosis, the administration of testosterone propionate improved the course of the disease by suppressing TGFβ1/Smad signaling; reducing the expression of type I and IV of collagen, fibronectin, TIMP1 and TIMP2b; and increasing the expression of MMP-2 and MMP-9, as well as activation of Nrf2-antioxidant response element (ARE) signaling. It was suggested that the protective effects of testosterone propionate in renal fibrosis could depend on the activation of Nrf2-ARE signaling, which reduces ROS production [118]. However, testosterone administration to ovariectomized rats induced podocyte apoptosis [119], which was associated with reduced activity of NO synthase, Akt protein kinase and decreased ERK/JNK phosphorylation. In a rat model of UUO, testosterone enhanced TNFα production and profibrotic and proapoptotic signaling, leading to TF [120]. Androgen deprivation in patients with prostate cancer reduced the rate of tumor progression, but caused dyslipidemia, hyperglycemia, increased fat mass and increased the risk of AKI [121]. This is probably due to the fact that androgen deprivation reduces the level of estrogens, which have a protective effect on the kidney [122].

#### 3.5.2. Estradiol

In addition to testosterone, female sex hormones play an important role in the kidney tolerance to damage under the condition of AKI, hypertension, diabetes and other diseases that affect kidney function. Estradiol (17β-estradiol) has a protective effect on the kidney and suppresses fibrosis, in contrast to testosterone. Estradiol did not cause podocyte apoptosis preceding glomerulosclerosis, which was shown in the estrogen receptor (ER)-knockout mouse model [119], reduced albuminuria, TF progression and glomerulosclerosis by suppressing the activity of MMP-2 and MMP-9 in diabetic nephropathy [123,124]. These effects can also be explained by the influence on the expression of TGFβ1 and collagen, which leads to synthesis modification and ECM degradation [108]. ERα activation reduced podocyte death, possibly due to stabilization of the mitochondrial membrane potential and activation of MAPK signaling pathways [125].

Despite an impressive amount of work about the positive effect of estradiol on kidney function, there is evidence of its negative impact as well. Lakzaei et al. reported an increase in ischemic damage to kidney tissues when estradiol was administered to ovariectomized rats compared to animals treated with losartan and Ang 1-7 in an ischemia-reperfusion model [126].

Estradiol is currently considered to be a potential hormonal nephroprotective [122], but it is necessary to clarify how to avoid possible negative consequences of using ER agonists in the treatment of kidney fibrosis.

#### 3.5.3. Progesterone

Progesterone is traditionally assigned the role of a hormone responsible for the initiation and maintenance of pregnancy; however, this steroid has a wide spectrum of action, including the regulation of the functioning of organs and systems that are not directly related to reproductive ones, such as the brain, mammary gland, cardiovascular and immune systems [127,128]. Progesterone realizes its effects through two types of receptors: nuclear and membrane. It is believed that most of its effects in humans and other mammals are mediated by nuclear receptors, which are transcription factors, upon activation of which slow genomic effects are mainly realized [129]. Extragenomic effects are mediated by both nuclear and membrane progesterone receptors [129,130,131].

The role of progesterone in kidney physiology is not fully understood yet. Its receptors are expressed in different nephron segments in humans and other mammals: in the proximal and distal tubules and glomerulus [132,133]. It was recently shown that the expression of membrane progesterone receptors PAQR5 was suppressed by TGFβ1 and associated with a poor survival outcome in renal clear cell carcinoma. In a model of diabetic nephropathy in ovariectomized female rats, progesterone administration suppressed the progression of kidney fibrosis by reducing the expression of fibronectin, TGFβ1 and AT1 receptors [134]. There is also evidence of a positive effect of synthetic progestins on the course of kidney fibrosis [135]. A number of studies showed the antioxidant effect of progesterone: the progesterone administration to ovariectomized female rats with acute or chronic kidney damage caused a decrease in malondialdehyde serum levels (a marker of oxidative stress) [136], lactate dehydrogenase activity, lipid peroxidation and generation of superoxide anions [137,138]. An in vitro study of the progesterone effect on fibrogenesis showed that incubation of cardiac fibroblasts isolated from male rats with uremic cardiomyopathy and 5/6 nephrectomy with progesterone, in combination with the cardiotonic steroid marinobufagenin, was more effective in reducing type I collagen expression than incubation with marinobufagenin alone [139]. However, further study of the mechanisms of progesterone action on the kidney is required, as well as a gender comparison of its effects.

The scheme illustrating the influence of sex hormones on fibrosis development is shown in Figure 2.

## 4. Hormone Therapy of Fibrosis

Modern therapeutic approaches to hormonal correction of fibrosis can be divided into several strategies: blockade of profibrotic hormone signaling, use of profibrotic hormone receptor antagonists, the use of agonists of hormones with antifibrotic effects, inhibitors and stimulators of enzymes involved in hormone biosynthesis and a combination of these approaches (Figure 3).

As mentioned above, TGFβ1 signaling is considered to be the basis of fibrogenesis of the kidney and other organs; thus, it was proposed to block this signaling by using antibodies and inhibitors of serine-threonine kinases to suppress it [43]. Fresolimumab, which suppresses the synthesis of all three TGFβ1 isoforms but does not reduce proteinuria and high blood pressure in patients with primary focal segmental glomerulosclerosis [140], can be cited as an example of blockade using human monoclonal antibodies. Pirfenidone, which inhibits the synthesis of TGFβ1, is successfully used to treat pulmonary fibrosis but showed ambiguous effects in the treatment of kidney fibrosis [43,141]. Bone morphogenic protein 7 (BMP-7) is a TGFβ1 antagonist, so it also deserves close attention. BMP-7 is predominantly expressed in cortical and corticomedullary proximal tubular cells, distal convoluted tubules, collecting duct epithelia and glomerular podocytes [142]. This protein showed inhibition of the development of kidney fibrosis by suppressing the ECM deposition (collagen I and III [142]), inhibiting apoptosis of renal tubular cells [143] and suppressing TGFβ1 signaling itself, affecting miR-21/Smad7 [144] and Smad 2/3 pathway [142] in several preclinical studies.

One of the most promising strategies for the treatment of kidney fibrosis is the correction of RAAS function. The regulation of this system realizes through the use of renin inhibitors, AT1 antagonists, AT2 agonists, ACE1 and chymase inhibitors (an enzyme involved in the alternative Ang II synthesis pathway), as well as ACE2 stimulators [145,146]. Despite the significant antifibrotic effects of the aforementioned substances in vivo and in vitro, the use of AT1 and ACE1 blockers alone did not sufficiently suppress CKD progression, in contrast to their combination. The use of this combination has long been considered the gold standard of CKD therapy; however, the antifibrotic effect caused by these drugs does not lead to a complete restoration of kidney function [146]. Another variant of the combination of drugs blocking various parts of the RAAS was also used: the AT1 blocker losartan and the renin inhibitor aliskiren in patients with nephropathy and type 2 diabetes mellitus showed a nephroprotective effect, manifested in a significant alleviation in proteinuria [147]. Aliskiren showed a pronounced antifibrotic effect in a number of in vivo studies [148] and is expected to be used in clinical practice in the near future. Recently, a new approach for the treatment of CKD has been proposed—the use of a combination of the neprilysin inhibitor (sacubitril), which inhibits the cleavage of natriuretic peptides, and the AT1 antagonist valsartan (ARNI—angiotensin receptor AT1 and neprilysin inhibition). This combination of drugs showed a positive effect on the suppression of the development of TF and glomerulosclerosis in rats [149] and in clinical studies of acute renal pathologies treatment; however, it is necessary to confirm the positive effects in patients with CKD [150].

Substance C21, a synthetic AT2 agonist, showed antifibrotic and antithrombotic effects in animal models [145] and started to be introduced into clinical practice. C21 suppressed the proinflammatory response, the release of TNF-α and interleukine-6 during the acute phase of renal ischemia-reperfusion and stimulated anti-inflammatory responses in the delayed post-ischemic period due to the activation of T-regulatory cells and the release of interleukine-10 [60,145].

It should be noted the therapeutic strategy aimed at correcting the work of the protective branch of the RAAS associated with ACE2 activation, which promotes Ang 1-7 formation from Ang II. The use of recombinant ACE2 in clinical practice in healthy people significantly reduced the level of Ang II and increased Ang 1-7 content. How-ever, this clinical approach may be effective in the treatment of neglected diabetic nephropathy only, when severe proteinuria is observed due to significant damage of the glomerular filter, which allows recombinant ACE2 to enter the tubules [151].

Although Ang 1-7 is considered as an antifibrotic agent demonstrated beneficial effects in animal models, there were only few clinical observations, that limits its prac-tical use [152]. Thus, some studies have showed a positive effect of intrarenal admin-istration of Ang 1-7 in patients with renal artery stenosis due to fibromuscular dyspla-sia [153,154]. The full implementation of Ang 1-7 into clinical practice is also hindered by the peculiarities of its pharmacokinetics. This peptide has a very short half-life due to rapid cleavage by peptidases [152]. To avoid this problem synthetic MAS1 receptor agonists were developed, for example, AVE0991 [155], however it is has not yet been used in clinical practice. 

The use of abovementioned estradiol for the treatment of renal fibrosis currently requires further study due to the controversial results obtained during in vitro and in vivo experiments. Some clinical studies showed proteinuria reduction in patients with diabetes and hypertension [156] and slowing progression of CKD in postmenopausal women on estrogen hormone replacement therapy, which was attributed to an in-crease in renal NO production and a decrease in oxidative stress [157]. The selective estrogen receptor modulator tamoxifen used to treat breast cancer displayed an antifi-brotic effect in the treatment of retroperitoneal fibrosis, one of the frequent complica-tions of which is ureteral obstruction [158]. Fibrosis regression was shown in 70% of cases in a clinical study of 31 patients with retroperitoneal fibrosis, but with a number of side effects: pulmonary embolism (3.2%), deep vein thrombosis (3.2%), ovarian cyst (3.2%), and headache (9.7%). The issue of using tamoxifen in this case as an alternative to glucocorticoid therapy remains open [159,160].

Another therapeutic strategy in the treatment of CKD is the suppression of hy-perprolactinemia using dopamine receptor agonists (cabergoline, bromocriptine), which is a prolactostatin [107]. However, there is insufficient data about effectiveness of such therapy in patients on dialysis and with CKD at the moment [81].

Little is known about a potential of natriuretic peptides to reduce the progression of renal fibrosis, and existing data are conflicting. ANP infusion caused a vasodilatory effect, increased renal blood flow in patients with CKD and reduced the risk of con-trast-induced nephropathy after coronary angiography [161,162,163], but it could cause side effects such as bradycardia and hypotension [164]. On the contrary, the ANP pre-cursor did not cause a systemic vasodilator effect and can improve kidney function by induction of the nephroprotective prostaglandin E2 synthesis [165]. Synthetic ana-logues of ANP, carperitide and anaritide, did not possess a beneficial effect in this type of nephropathy [166,167], moreover anaritide enhanced proteinuria and natriuresis in patients with nephrotic syndrome [168]. Thus, there are a limited number of high-quality randomized placebo-controlled trials to draw a conclusion about the role of ANP in the treatment of CKD [169].

## 5. Conclusions

Hormones represent one of the important actors in both kidney fibrosis and the pathological conditions that precede it. Hormonal therapy in fibrosis has a number of advantages associated with the pleiotropic effect of hormones. Nephroprotective effects are realized not only at the systemic level but also in mitochondrial functions and ECM synthesis and generally improve a number of physiological parameters associated with CKD and fibrosis. Nevertheless, such systemic action has a downside since positive effects in some systems generate adverse effects in others. Therefore, although therapy aimed at the hormonal mechanisms of fibrosis development seems to be very promising, now it seems most appropriate to use it in combination with other antifibrotic and nephroprotective approaches.

## Figures and Tables

**Figure 1 life-12-00737-f001:**
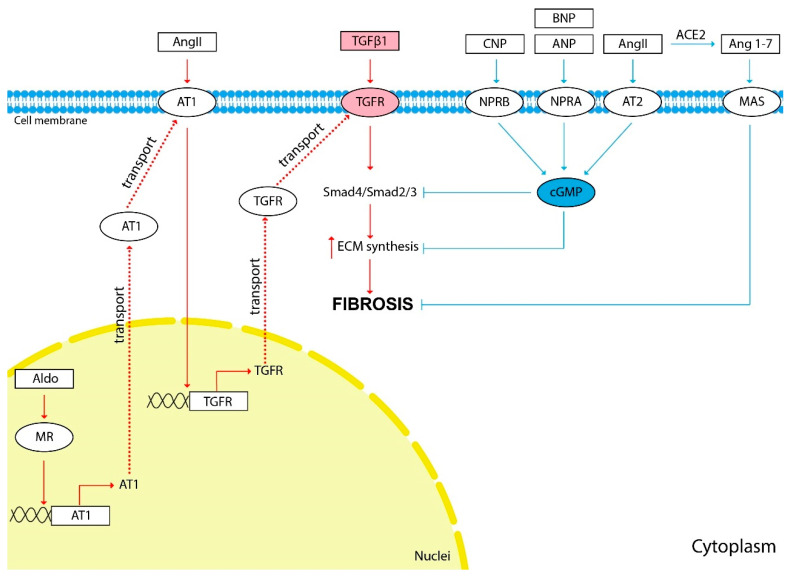
The influence of vasoactive hormones on kidney fibrosis development. The red arrows show the hormonal effect that stimulates fibrosis development, and the blue arrows show the inhibitory effect. TGFβ1 induces the synthesis of ECM components and is the main cytokine involved in kidney fibrosis development. Most of the hormones discussed in this review affect the progression of fibrosis indirectly through TGFβ1 stimulation. Pathological activation of the RAAS causes activation of ECM synthesis and kidney fibrosis development. TGFβ1 is an important mediator of the effects of Ang II and Aldo on TF development. Ang II stimulates the expression of the TGFβ1 receptor (TGFR) through AT1. Aldo stimulates the expression of AT1 through its nuclear receptors. Natriuretic peptides ANP, BNP, CNP and Ang II through AT2 inhibit TGFβ1 signaling and ECM synthesis through cGMP second messenger. Ang 1-7 is generated from Ang II by ACE2. Ang 1-7 through the MAS membrane receptor demonstrate effects similar to those of Ang II through the AT2—anti-inflammatory, antifibrotic and vasodilating.

**Figure 2 life-12-00737-f002:**
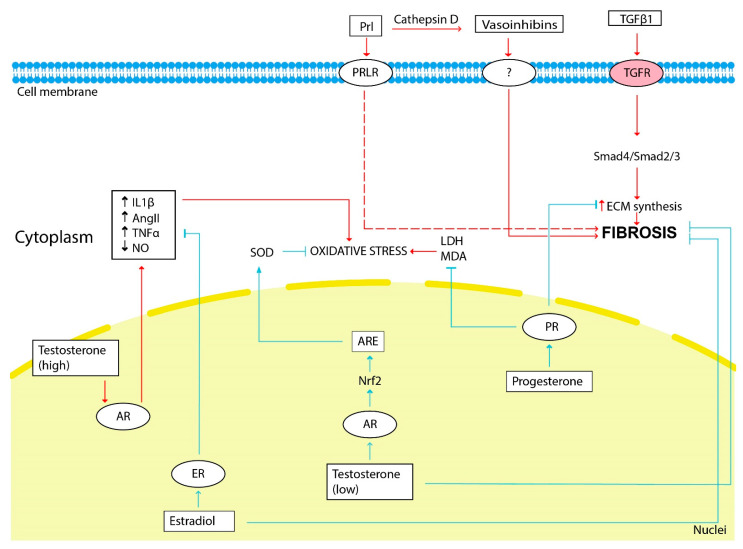
The influence of sex hormones on kidney fibrosis development. The red arrows show the hormonal effect that stimulates fibrosis development, and the blue arrows show the inhibitory effect. Estradiol, testosterone (low doses) and progesterone through its receptors (ER, AR and PR, respectively) inhibit ECM synthesis, fibrosis progression and oxidative stress. Prolactin and vasoinhibins generated from prolactin by cathepsin D stimulate fibrosis progression. Renal oxidative stress is stimulated by testosterone (high doses). LDH—lactate dehydrogenase, MDA—malondialdehyde.

**Figure 3 life-12-00737-f003:**
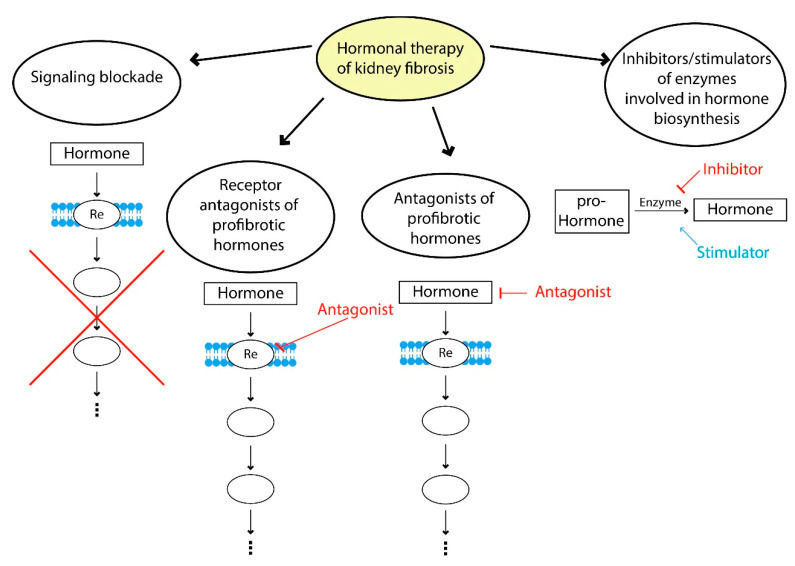
Hormone therapy of fibrosis. Examples of signaling blockade: suppression of TGFβ1 signaling by bone morphogenic protein 7; use of receptor antagonists of profibrotic hormones—AT1 receptors antagonists (valsartan, losartan); examples of antagonists of profibrotic hormones—suppression of hyperprolactinemia using dopamine receptor agonists (cabergoline, bromocriptine); inhibitors and stimulators of enzymes involved in the hormone biosynthesis—renin inhibitors (aliskiren), neprilysin inhibitors (sacubitril), ACE2 stimulators.

## Data Availability

The data that support the findings of this study are available from the corresponding author upon reasonable request.

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
