# Peer review of "Hormonal Regulation of Renal Fibrosis"

_life, 2022, doi:10.3390/life12050737_

Round 1

Reviewer 1 Report

Very Good, well written papier

Author Response

We thank the reviewer for the high appreciation of our work.

Reviewer 2 Report

A fairly comprehensive review of the hormonal involvement in kidney fibrosis. It is well referenced and has provided a balanced viewpoint.  It does not mention thefew instances where  need for glutamate metabolism when lipolysis is suppressed, as they point out glycolysis cannot sustain the energy requirements? There is a need for another nutrient to maintain the production of ROS as Glycolysis is unlikely to be able to increase ROS.  There a few glitches in the english expression, nothing serious e.g. line 104 refers to glycolysis activity not glycolytic. A reference to the point made in lines 133 to 135 needs to be added. a possible paper is Wang et al. Cell Death and Disease (2020) 11:29
https://doi.org/10.1038/s41419-019-2218-5. Line 396"refer Prl to uremic toxins" needs to be made clearer.  

Figure 3 needs some more information added to be useful as the text describes some actions not indicated in the figure. BNP-7 is introduced in the text below figure 3 but barely rates a mention elsewhere in the body of the document, What is its Role?

The balance between synthesis and degradation of fibrosis needs to be expanded as a concept responsible for the uncontrolled fibrosis.  It concentrates on MMP-7 but other studies have shown the MMP-9 and 2 are activated  and MMP-7 has a role in  innate immunity , is this a result of gut biome changes? in response to inflammation ? It is also a biomarker for kidney fibrosis in the serum  and as such may have a signaling function whereas the other MMP seem to be more localised in effect. NGAL does not get mentioned but is also a marker and indicator of the fibrotic condition, Why was this not included?

Author Response

Please, find our replies in the attached file.

Reviewer 3 Report

I congratulate the authors for a well written review. I enjoyed reading the review and learnt several aspects related to kidney fibrosis and its regulation by hormones.  

Author Response

We thank the reviewer for the nice comments and are glad that the data and our analysis were useful to him/her.